# Variational Bayesian Optimistic Sampling

**Brendan O'Donoghue**
DeepMind
bodonoghue@deepmind.com

**Tor Lattimore**
DeepMind
lattimore@deepmind.com

## Abstract

We consider online sequential decision problems where an agent must balance exploration and exploitation. We derive a set of Bayesian 'optimistic' policies which, in the stochastic multi-armed bandit case, includes the Thompson sampling policy. We provide a new analysis showing that any algorithm producing policies in the optimistic set enjoys $\tilde{O}(\sqrt{AT})$ Bayesian regret for a problem with $A$ actions after $T$ rounds. We extend the regret analysis for optimistic policies to bilinear saddle-point problems which include zero-sum matrix games and constrained bandits as special cases. In this case we show that Thompson sampling can produce policies outside of the optimistic set and suffer linear regret in some instances. Finding a policy inside the optimistic set amounts to solving a convex optimization problem and we call the resulting algorithm 'variational Bayesian optimistic sampling' (VBOS). The procedure works for any posteriors, *i.e.*, it does not require the posterior to have any special properties, such as log-concavity, unimodality, or smoothness. The variational view of the problem has many useful properties, including the ability to tune the exploration-exploitation tradeoff, add regularization, incorporate constraints, and linearly parameterize the policy.

## 1 Introduction

In this manuscript we consider online learning, in particular the multi-armed stochastic bandit problem [19] and its extension to bilinear saddle-point problems. Thompson sampling (TS) is a well-known Bayesian algorithm to tackle this problem [37, 35]. At each iteration TS samples an action according to the posterior probability that it is the optimal action. However, it never computes these probabilities, instead TS samples from the posterior of each arm and then acts greedily with respect to that sample, which is an *implicit* sample from the posterior probability of optimality. TS enjoys strong theoretical guarantees [1, 35] as well as excellent practical performance [6]. Moreover, it has an elegant information-theoretic interpretation of an agent that is making a tradeoff between information accumulation and regret [34]. That being said it does have several weaknesses. Firstly, TS requires samples from the true posterior, which may be intractable for some problems. It is known that TS with even small error in its posterior samples can suffer linear regret [32]. In many cases of interest the posterior distribution is highly intractable to sample from and running MCMC chains, one for each arm, can be wasteful and slow and still not yield precise enough samples for good performance, though recent work has attempted to improve this situation [22]. Moreover, TS is unable to take into account other preferences, such as constraints or the presence of other agents. In particular, it was recently shown that TS can suffer linear regret in two-player zero sum games [27] and we shall demonstrate the same phenomenon in this work empirically for constrained bandits. Finally, when performing TS the user never has direct access to the posterior probability of optimality, only samples from that distribution (though the distribution can be estimated by taking many samples). In some practical applications having access to the policy is desirable, for example to ensure safety constraints or to allocate budgets.

35th Conference on Neural Information Processing Systems (NeurIPS 2021).

In this paper we present 'variational Bayesian optimistic sampling' (VBOS), a new Bayesian approach to online learning. At every step the VBOS algorithm solves a convex optimization problem over the simplex. The solution to this problem is a policy that satisfies a particular 'optimism' condition. The problem is always convex, no matter what form the posteriors take. The only requirement is the ability to compute or estimate the cumulant generating function of the posteriors. VBOS and TS can be considered as part of the same family of algorithms since, in the bandit case, they both produce policies inside the optimistic set. That being said, VBOS has several advantages over TS. First, in some cases estimating or bounding the cumulant generating function may be easier than sampling from the posterior. Second, incorporating additional requirements such as constraints, opponents or tuning the exploration exploitation tradeoff is easy in VBOS - just change the optimization problem. Finally, when using VBOS we always have direct access to the policy, rather than just implicit samples from it, as well as an explicit upper bound on the expected value of the problem, which TS does not give us. Here we list the contributions of this work.

## 1.1 Contributions

1. We derive a set of 'optimistic' Bayesian policies for the stochastic multi-armed bandit that satisfy a particular optimism inequality and which includes the TS policy.

2. We present a new, simple proof of a $\tilde{O}(\sqrt{AT})$ regret-bound for a bandit with $A$ arms after $T$ timesteps that covers any policy in the optimistic set ($\tilde{O}$ ignores logarithmic factors).

3. We derive *variational Bayesian optimistic sampling* (VBOS), which is a policy in the optimistic set that can be computed by solving a convex optimization problem.

4. We show that, unlike TS, the variational formulation easily handles more complicated problems, such as bilinear saddle-point problems which include as special cases zero-sum two player games and constrained bandits.

## 1.2 Preliminaries

A stochastic multi-armed bandit problem is a sequential learning problem in which an agent interacts with an environment in order to maximize its total cumulative reward. Initially the agent does not know how the rewards are distributed and must learn about them from experience. In each round $t \in \mathbb{N}$ the agent selects an action $a_t \in \{1, \ldots, A\}$ and receives a reward $r_t = \mu_{a_t} + \eta_{a_t}^t$ where $\eta_{a_t}^t \in \mathbb{R}$ is zero-mean noise associated with action $a_t$ at time $t$. We define $\mathcal{F}_t = (a_1, r_1, \ldots, a_{t-1}, r_{t-1})$ to be the sequence of actions and rewards observed by the agent prior to round $t$, and as shorthand we shall use the notation $\mathbb{E}^t(\cdot) = \mathbb{E}(\cdot \mid \mathcal{F}_t)$.

An algorithm alg is a measurable mapping that takes the entire history at time $t$ and produces a probability distribution, the *policy*, from which the agent samples its action. In order to assess the quality of an algorithm alg we consider the *regret*, or shortfall in cumulative rewards relative to the optimal value,

$$\text{Regret}(\mu, \text{alg}, T) = \mathbb{E}_{\eta, \text{alg}} \left[ \sum_{t=1}^{T} \max_i \mu_i - r_t \right], \tag{1}$$

where $\mathbb{E}_{\eta, \text{alg}}$ denotes the expectation with respect to the noise process $\eta$ and any randomness in the algorithm. This quantity (1) depends on the unknown expected reward $\mu \in \mathbb{R}^A$, which is fixed at the start of play and kept the same throughout. To assess the quality of learning algorithms designed to work across some *family* $\mathcal{M}$ of problems we define

$$\text{BayesRegret}(\phi, \text{alg}, T) = \mathbb{E}_{\mu \sim \phi} \text{Regret}(\mu, \text{alg}, T) \tag{2}$$

where $\phi$ is a prior probability measure over $\mu \in \mathcal{M}$ that assigns relative importance to each problem instance and which we assume is known to the algorithm.

## 2 Thompson sampling and a variational approach

The algorithm that minimizes the Bayesian regret (2) is called the *Bayes-optimal* policy, but it is believed to be intractable to compute in most cases [10]. With that in mind we would like an algorithm that is computationally tractable and still guarantees a good bound on the Bayesian regret.

TS is a Bayesian algorithm that at each time step samples possible values of each arm from their respective posteriors and acts greedily with respect to those samples; see Algorithm 1. This simple strategy provides a principled approach to the exploration-exploitation tradeoff in the stochastic multi-armed bandit problem. In this section we derive a set of 'optimistic' policies, which includes the TS policy, and we shall show that any algorithm that produces policies in this set satisfies a guaranteed Bayesian regret bound. For brevity we defer all proofs to the appendices.

## 2.1 Bounding the conditional expectation

The Bayesian regret (2) depends on the expectation of the maximum over a collection of random variables. Here we shall derive an upper bound on this quantity, starting with a generic upper bound on the conditional expectation.

If $X : \Omega \to \mathbb{R}^d$ is a random variable in $L_1^d$ (*i.e.*, $\mathbb{E}|X_i| < \infty$ for each $i = 1, \ldots, d$) then we denote by $\Psi_X : \mathbb{R}^d \to \mathbb{R} \cup \{\infty\}$ the cumulant generating function of $X - \mathbb{E}X$, *i.e.*,

$$\Psi_X(\beta) = \log \mathbb{E} \exp(\beta^\top (X - \mathbb{E}X)).$$

This function is always convex, and we shall assume throughout this manuscript that it is also closed and proper for any random variables we encounter. With this definition we can present a bound on the conditional expectation.

**Lemma 1.** *Let $X : \Omega \to \mathbb{R}$ be a random variable on $(\Omega, \mathcal{F}, \mathbb{P})$ satisfying $X \in L_1$, and let $A \in \mathcal{F}$ be an event with $\mathbb{P}(A) > 0$. Then, for any $\tau \geq 0$*

$$\mathbb{E}[X|A] \leq \mathbb{E}X + \tau \Psi_X(1/\tau) - \tau \log \mathbb{P}(A). \tag{3}$$

The right hand side of (3) involves the perspective of $\Psi_X$ and so is convex in $\tau$ [4] and we can minimize over $\tau \geq 0$ to find the tightest bound, which we do next. Recall that the *convex conjugate* of function $f : \mathbb{R}^d \to \mathbb{R}$ is denoted $f^* : \mathbb{R}^d \to \mathbb{R}$ and is given by $f^*(y) = \sup_x \left( x^\top y - f(x) \right)$ [4].

**Theorem 1.** *Let $X : \Omega \to \mathbb{R}$ be a random variable such that the interior of the domain of $\Psi_X$ is non-empty, then under the same assumptions as Lemma 1 we have* [1]

$$\mathbb{E}[X|A] \leq \mathbb{E}X + (\Psi_X^*)^{-1}(-\log \mathbb{P}(A)).$$

For example, if $X \sim \mathcal{N}(0, \sigma^2)$, then for any event $A$ we can bound the conditional expectation as $\mathbb{E}[X|A] \leq \sigma\sqrt{-2\log \mathbb{P}(A)}$, no matter how $X$ and $A$ are related. This inequality has clear connections to Kullback's inequality [8] and the Donsker-Varadhan variational representation of KL-divergence [11, Thm. 3.2]. The function $\Psi^*$ is referred to as the *Cramér function* or the *rate function* in large deviations theory [39]. The inverse of the rate function comes up in several contexts, such as queuing theory and calculating optimal insurance premiums [21]. Cramér's theorem tells us that the probability of a large deviation of the empirical average of a collection of IID random variables decays exponentially with more data, and that the decay constant is the rate function. It is no surprise then that the regret bound we derive will depend on the inverse of the rate function, *i.e.*, when the tails decay faster we incur less regret.

## 2.2 The optimistic set

Using Theorem 1 we next derive the maximal inequality we shall use to bound the regret.

**Lemma 2.** *Let $\mu : \Omega \to \mathbb{R}^A$, $\mu \in L_1^A$, be a random variable, let $i^\star = \operatorname{argmax}_i \mu_i$ (ties broken arbitrarily) and denote by $\Psi_i := \Psi_{\mu_i}$, then*

$$\mathbb{E} \max_i \mu_i \leq \sum_{i=1}^A \mathbb{P}(i^\star = i) \left( \mathbb{E}\mu_i + (\Psi_i^*)^{-1}(-\log \mathbb{P}(i^\star = i)) \right).$$

Note that Lemma 2 makes no assumption of independence of each entry of $\mu$, the bound holds even in the case of dependency.

---

[1] If $\Psi_X = 0$, then we take $(\Psi_X^*)^{-1} = 0$.

In the context of online learning $\pi_i^{\mathrm{TS}} = \mathbb{P}(i^\star = i)$ is exactly the TS policy when $\mu_i$ have the laws of the posterior. For ease of notation we introduce the 'optimism' map $\mathcal{G}_\phi^t : \Delta_A \to \mathbb{R}$ which for a random variable $\mu : \Omega \to \mathbb{R}^A$ distributed according to $\phi(\cdot \mid \mathcal{F}_t)$ is given by

$$\mathcal{G}_\phi^t(\pi) := \sum_{i=1}^A \pi_i \left( \mathbb{E}^t \mu_i + (\Psi_i^{t*})^{-1}(-\log \pi_i) \right), \tag{4}$$

where $\Delta_A$ denotes the probability simplex of dimension $A - 1$ and $\Psi_i^t := \Psi_{\mu_i | \mathcal{F}_t}$. Note that $\mathcal{G}_\phi^t$ is concave since $\Psi_i^t$ is convex [4]. Moreover, if the random variables are non-degenerate then $\mathcal{G}_\phi^t$ is differentiable and the gradient can be computed using Danskin's theorem [7].

With this notation we can write $\mathbb{E}^t \max_i \mu_i \leq \mathcal{G}_\phi^t(\pi^{\mathrm{TS}})$. This brings us to the 'optimistic' set of probability distributions, which corresponds to the set of policies that satisfy that same inequality.

**Definition 1.** *Let* $\mu : \Omega \to \mathbb{R}^A$ *be a random variable distributed according to* $\phi(\cdot \mid \mathcal{F}_t)$*, then we define the optimistic set as*

$$\mathcal{P}_\phi^t := \{\pi \in \Delta_A \mid \mathbb{E}^t \max_i \mu_i \leq \mathcal{G}_\phi^t(\pi)\}.$$

Immediately we have two facts about the set $\mathcal{P}_\phi^t$. First, it is non-empty since $\pi^{\mathrm{TS}} \in \mathcal{P}_\phi^t$, and second, it is a convex set since the optimism map $\mathcal{G}_\phi^t$ is concave.

## 2.3 Variational Bayesian optimistic sampling

Since the TS policy is in $\mathcal{P}_\phi^t$, it implies that we can maximize the bound and also obtain a policy in $\mathcal{P}_\phi^t$, since

$$\mathbb{E}^t \max_i \mu_i \leq \mathcal{G}_\phi^t(\pi^{\mathrm{TS}}) \leq \max_{\pi \in \Delta_A} \mathcal{G}_\phi^t(\pi). \tag{5}$$

We refer to the policy that maximizes $\mathcal{G}_\phi^t$ as the 'variational Bayesian optimistic sampling' policy (VBOS) and present the VBOS procedure as Algorithm 2. VBOS produces a policy $\pi^t \in \mathcal{P}_\phi^t$ at each round by construction. The maximum is guaranteed to exist since $\Delta_A$ is compact and $\mathcal{G}_\phi^t$ is continuous. The problem of maximizing $\mathcal{G}_\phi^t$ is a concave maximization problem, and is thus computationally tractable so long as each $\Psi_i^t$ is readily accessible. This holds *no matter what form the posterior distribution takes*. This means we don't rely on any properties like log-concavity, unimodality, or smoothness of the distribution for tractability of the optimization problem.

As an example, consider the case where each $\mu_i \sim \mathcal{N}(0, \sigma^2)$ independently. In this instance we recover the familiar bound of $\mathbb{E} \max_i \mu_i \leq \mathcal{G}_\phi(\pi^{\mathrm{TS}}) = \max_{\pi \in \Delta_A} \mathcal{G}_\phi(\pi) = \sigma\sqrt{2\log A}$. The variational form allows us to compute a bound in the case of differently distributed random variables.

---

**Algorithm 1** TS for bandits

  **for** round $t = 1, 2, \ldots, T$ **do**
    sample $\hat{\mu}^t \sim \phi(\cdot \mid \mathcal{F}_t)$
    choose $a_t \in \mathrm{argmax}_i \hat{\mu}_i^t$
  **end for**

---

**Algorithm 2** VBOS for bandits

  **for** round $t = 1, 2, \ldots, T$ **do**
    compute $\pi^t = \mathrm{argmax}_{\pi \in \Delta_A} \mathcal{G}_\phi^t(\pi)$
    sample $a_t \sim \pi^t$
  **end for**

---

## 2.4 Regret analysis

Now we shall present regret bounds for algorithms that produce policies in the optimistic set. These bounds cover both TS and VBOS.

**Lemma 3.** *Let* alg *produce any sequence of policies* $\pi^t$*,* $t = 1, \ldots, T$*, that satisfy* $\pi^t \in \mathcal{P}_\phi^t$*, then*

$$\mathrm{BayesRegret}(\phi, \mathrm{alg}, T) \leq \mathbb{E} \sum_{t=1}^T \sum_{i=1}^A \pi_i^t (\Psi_i^{t*})^{-1}(-\log \pi_i^t).$$

This bound is generic in that it holds for *any* set of posteriors and *any* algorithm producing policies in $\mathcal{P}_\phi^t$. The regret depends on the inverse rate functions of the posteriors, no matter how complicated

those posteriors may be, and faster concentrating posteriors, *i.e.*, larger rate functions, will have lower regret. Loosely speaking, as more data accumulates the posteriors concentrate towards the true value (under benign conditions [38, Ch. 10.2]), implying $(\Psi_i^{t*})^{-1} \to 0$. By carefully examining the rate of concentration of the posteriors we can derive concrete regret bounds in specific cases, such as the standard sub-Gaussian reward noise case, which we do next.

**Theorem 2.** *Let* alg *produce any sequence of policies* $\pi^t$, $t = 1, \ldots, T$, *that satisfy* $\pi^t \in \mathcal{P}_\phi^t$ *and assume that both the prior and reward noise are* 1*-sub-Gaussian for each arm, then*

$$\text{BayesRegret}(\phi, \text{alg}, T) \leq \sqrt{2AT \log A(1 + \log T)} = \tilde{O}(\sqrt{AT}).$$

The assumption that the prior is 1-sub-Gaussian is not really necessary and can be removed if, for example, we assume the algorithm pulls each arm once at the start of the procedure. Theorem 2 bounds the regret of *both* TS and VBOS, as well as any other algorithm producing optimistic policies. The point of this theorem is not to provide the tightest regret bound but to provide some insight into the question of how close to the true TS policy another policy has to be in order to achieve similar (in particular sub-linear) regret. We have provided one answer - any policy in the optimistic set $\mathcal{P}_\phi^t$ for all $t$, is close enough. However, this set contracts as more data accumulates (as the posteriors concentrate) so it becomes harder to find good approximating policies. That being said, the VBOS policy is always sufficiently close because it is always in the optimistic set. In Figure 1 we show the progression of a single realization of the set $\mathcal{P}_\phi^t$ over time for a three-armed Gaussian bandit with Gaussian prior and where the policy being followed is TS [13]. Initially, $\mathcal{P}_\phi^t$ is quite wide, but gradually as more data accrues it concentrates toward the optimal arm vertex of the simplex.

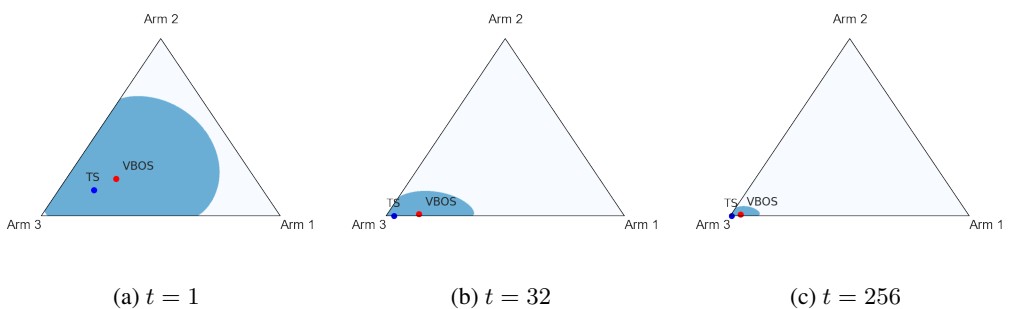

(a) $t = 1$             (b) $t = 32$             (c) $t = 256$

Figure 1: Policy simplex for a three-armed bandit. The shaded area is the optimistic set $\mathcal{P}_\phi^t$. The blue dot is the TS policy and the red dot is the VBOS policy.

## 3 Connections and interpretations of VBOS

**Exploration-exploitation tradeoff.** Examining the problem that is solved to determine the VBOS policy we can break it into two components

$$\mathcal{G}_\phi^t(\pi) = \underbrace{\sum_{i=1}^A \pi_i \mathbb{E}^t \mu_i}_{\text{exploit}} + \underbrace{\sum_{i=1}^A \pi_i (\Psi_i^{t*})^{-1}(-\log \pi_i)}_{\text{explore}},$$

the first term of which includes the expected reward (under the posterior) from pulling each arm and is the agent's current understanding of the world, and the second term consists only of the uncertainty in each arm. The policy that maximizes this function is balancing this exploration and exploitation tradeoff, with a tradeoff parameter of one. In some practical instances it may be beneficial to use a different tradeoff parameter other than one, *e.g.*, when there are a larger number of arms than the total number of rounds to play. In this case VBOS can easily be tuned to be more or less exploring as needed.

**Upper confidence bounds.** Here we show a connection between VBOS and upper confidence bound (UCB) methods [18, 2]. The Chernoff inequality tells us that the tail probability of random variable $\mu : \Omega \to \mathbb{R}$ can be bounded as $\mathbb{P}(\mu - \mathbb{E}\mu > \epsilon) \leq \exp(-\Psi_\mu^*(\epsilon))$, for any $\epsilon \geq 0$. UCB algorithms build upper confidence intervals which contain the true value of the random variable with high-probability [18, 2, 16]. If we want the interval to contain the true value of $\mu$ with probability at least $1 - \delta$ it requires setting $\epsilon = (\Psi_\mu^*)^{-1}(-\log \delta)$, yielding $\mathbb{E}\mu + (\Psi_\mu^*)^{-1}(-\log \delta) \geq \mu$ with high probability. UCB approaches to the stochastic multi-armed bandit typically choose the largest upper confidence bound (the 'optimism in the face of uncertainty' approach), that is, at time $t$ they select action

$$a_t \in \operatorname*{argmax}_i (\mathbb{E}^t \mu_i + (\Psi_i^{t*})^{-1}(-\log \delta_t)),$$

for some schedule $\delta_t$. A lot of work goes into tuning $\delta_t$ for various applications. By contrast, VBOS samples from policy

$$a_t \sim \operatorname*{argmax}_{\pi \in \Delta_A} \sum_{i=1}^{A} \pi_i \left( \mathbb{E}^t \mu_i + (\Psi_i^{t*})^{-1}(-\log \pi_i) \right).$$

It is clear that VBOS is building upper confidence bounds on the values of each arm that depend on the value of the policy, since if we denote $\pi^\star$ the maximizer to the above problem, then $\mathbb{P}^t(\mathbb{E}^t \mu_i + (\Psi_i^{t*})^{-1}(-\log \pi_i^\star) < \mu_i) \leq \pi_i^\star$. This procedure gives higher 'bonuses' to arms that end up with lower probability of being pulled, and lower bonuses to arms with high probability of being pulled, which is similar to what is done in more sophisticated UCB algorithms such as KL-UCB+ [17, 9, 14]. This is desirable because if the mean of an arm is low then there is a chance this is simply because we have under-estimated it and perhaps we should pay more attention to that arm. In this light VBOS can be interpreted as a stochastic Bayesian UCB approach [16]. One advantage is that we do not have to tune a schedule $\delta_t$, like TS our algorithm is randomized and *stationary*, *i.e.*, it does not depend on how many time-steps have elapsed other than implicitly through the posteriors.

**K-learning.** K-learning is an algorithm originally developed for efficient reinforcement learning in MDPs where it enjoys a similar regret bound to TS [23, 30, 31]. For bandits, the agent policy at time $t$ is given by

$$\tau_t^\star = \operatorname*{argmin}_{\tau \geq 0} \left( \tau \log \sum_{i=1}^{A} \exp(\mathbb{E}^t \mu_i / \tau + \Psi_i^t(1/\tau)) \right), \quad \pi_i^t \propto \exp(\mathbb{E}^t \mu_i / \tau_t^\star + \Psi_i^t(1/\tau_t^\star)). \quad (6)$$

It can be shown that this is equivalent to VBOS when the 'temperature' parameters $\tau_i$ that we optimize over are constrained to be equal for each arm. Although K-learning and VBOS have essentially the same regret bound, this can make a large difference in practice. This connection also suggests that recent reinforcement learning algorithms that rely on the log-sum-exp operator, the 'soft' max family [12, 20, 28, 33], are also in some sense approximating the VBOS policy, although as made clear in [29] these approaches do not correctly model the uncertainty and so will suffer linear regret in general.

## 4 Bilinear saddle-point problems

In this section we extend the above analysis to constrained bilinear saddle-point problems, which include as special cases zero-sum two player games and constrained multi-armed bandits, as well as standard stochastic multi-armed bandits. Equipped with the tools from the previous sections we shall formulate a convex optimization problem that when solved yields the VBOS policy at each round, and prove that in two special cases this strategy yields a sub-linear regret bound. It is known, and we shall demonstrate empirically, that TS (Algorithm 3) can fail, *i.e.*, suffer linear regret for some instances in this more general regime [27, §3.2]. Consequently, this is the first setting where the variational approach is substantially more powerful than TS.

A finite dimensional, constrained, bilinear saddle-point problem is defined by a matrix $R \in \mathbb{R}^{m \times A}$ and two non-empty convex sets $\Pi \subseteq \mathbb{R}^A$, $\Lambda \subseteq \mathbb{R}^m$ which are the feasible sets for the maximizer and minimizer respectively. Here we shall assume that the agent is the maximizer, *i.e.*, that $\Pi = \Delta_A$, and that the agent is uncertain about the entries of $R$ and must learn about them from experience. The saddle-point problem has optimal value given by

$$V_{R,\Lambda}^\star := \min_{\lambda \in \Lambda} \max_{\pi \in \Delta_A} \lambda^\top R \pi = \max_{\pi \in \Delta_A} \min_{\lambda \in \Lambda} \lambda^\top R \pi \quad (7)$$

where we can swap the order of the min-max due to the minimax theorem [40]. Unlike the stochastic multi-armed bandit case the maximizing policy in the bilinear saddle-point is typically stochastic, and as such the agent will return a policy rather than a single action at each time-step. This is to allow the case, for example, an opponent who gets to know the agent policy in advance of selecting their own action. We shall consider the case where at each time period the agent produces policy $\pi_t \in \Delta_A$ over columns of $R$ from which action $a_t \in \{1, \ldots, A\}$ is sampled, and receives reward $r_t \in \mathbb{R}$ and observation $o_t \in \mathcal{O}$ where either the reward, observation, or both are corrupted by noise $\eta^t$. The exact reward, noise, observation, and opponent space $\Lambda$ will depend on the problem setup. For instance, the vanilla stochastic multi-armed bandit is a special case of this problem where $\Lambda$ is a singleton and the agent receives bandit feedback associated with the action sampled from its policy (*i.e.*, $o_t = r_t = R_{a_t} + \eta^t_{a_t}$). The information available to the agent at the start of round $t$ is given by $\mathcal{F}_t = \{a_1, o_1, \ldots, a_{t-1}, o_{t-1}\}$. In this setup we define the Bayesian regret to be

$$\mathrm{BayesRegret}(\phi, \mathrm{alg}, T) = \mathbb{E}_{R \sim \phi} \left( \mathbb{E}_{\eta, \mathrm{alg}} \sum_{t=1}^{T} (V_{R,\Lambda}^\star - r_t) \right).$$

Analogously to the stochastic multi-armed bandit case we can define the optimism map and the optimistic set. Let $R_j : \Omega \to \mathbb{R}^m$, $j = 1, \ldots, A$, denote the random variable corresponding to the $j$th column of $R$, denote by $\Psi_j^t : \mathbb{R}^m \to \mathbb{R}$ the cumulant generating function of $R_j - \mathbb{E}^t R_j$, then assuming that $R_j \in L_1^m$ we define the optimism map in this case to be

$$\mathcal{G}_{\phi,\Lambda}^t(\pi) \coloneqq \min_{\lambda \in \Lambda, \tau \geq 0} \sum_{j=1}^{A} \pi_j (\lambda^\top \mathbb{E}^t R_j + \tau_j \Psi_j^t(\lambda/\tau_j) - \tau_j \log \pi_j) \tag{8}$$

and the optimistic set to be

$$\mathcal{P}_{\phi,\Lambda}^t \coloneqq \{\pi \in \Delta_A \mid \mathbb{E}^t V_{R,\Lambda}^\star \leq \mathcal{G}_{\phi,\Lambda}^t(\pi)\}. \tag{9}$$

With these we can bound $\mathbb{E}^t V_{R,\Lambda}^\star$ using the techniques we have previously developed.

**Lemma 4.** *Assuming that $R \in L_1^{m \times A}$ we have*

$$\mathbb{E}^t V_{R,\Lambda}^\star \leq \max_{\pi \in \Delta_A} \mathcal{G}_{\phi,\Lambda}^t(\pi). \tag{10}$$

This lemma motivates the VBOS algorithm for bilinear saddle-points, presented as Algorithm 4. VBOS produces policies $\pi^t \in \mathcal{P}_{\phi,\Lambda}^t$ for all $t$ by construction. On the other hand, in some cases TS can produce policies $\pi^t \notin \mathcal{P}_{\phi,\Lambda}^t$, which we shall show by counter-example in the sequel.

At first glance maximizing $\mathcal{G}_{\phi,\Lambda}^t$ might seem challenging, but we can convert it into a convex minimization problem by maximizing over $\pi$ to yield dual convex optimization problem:

$$\begin{aligned}
\text{minimize} \quad & V + \sum_{j=1}^{A} \tau_j \exp(s_j/\tau_j) \\
\text{subject to} \quad & s_j \geq \lambda^\top \mathbb{E}^t R_j + \tau_j \Psi_j^t(\lambda/\tau_j) - V - \tau_j, \quad j = 1, \ldots, A \\
& \tau \geq 0, \lambda \in \Lambda,
\end{aligned} \tag{11}$$

over variables $\tau \in \mathbb{R}^A, \lambda \in \mathbb{R}^m, s \in \mathbb{R}^A, V \in \mathbb{R}$. The VBOS distribution can be recovered using $\pi_i^\star = \exp(s_i^\star/\tau_i^\star)$. Problem (11) is an *exponential cone program* and can be solved efficiently using modern methods [25, 26, 36, 24].

---

**Algorithm 3** TS for saddle-points

> **for** round $t = 1, 2, \ldots, T$ **do**
>     sample $\hat{R}^t \sim \phi(\cdot \mid \mathcal{F}_t)$
>     set $\pi^t = \underset{\pi \in \Delta_A}{\mathrm{argmax}} \underset{\lambda \in \Lambda}{\min} \lambda^\top \hat{R}^t \pi$
> **end for**

**Algorithm 4** VBOS for saddle-points

> **for** round $t = 1, 2, \ldots, T$ **do**
>     set $\pi^t = \mathrm{argmax}\, \mathcal{G}_{\phi,\Lambda}^t(\pi)$
> **end for**

---

## 4.1 Zero-sum two-player games

Consider the problem of an agent playing a two-player zero-sum matrix game against an opponent. In this setting the agent 'column player' selects a column index $j \in \{1, \ldots, A\}$ and the 'row player'

selects a row index $i \in \{1, \ldots, m\}$, possibly with knowledge about the policy the agent is using to select the column. Note that in this case $\Lambda = \Delta_m$. These choices are revealed to the players and the row player makes a payment of $r_t = R_{ij} + \eta_{ij}^t$ to the column player, where $\eta_{ij}^t$ is a zero-mean noise term associated with actions $i, j$ at time $t$, and the observation the agents receive is the reward and the actions of both players. The optimal strategies $(\pi^\star, \lambda^\star)$ are a Nash equilibrium for the game [40, 5]. We shall assume that the agent we control is the column player, and that we have no control over the actions of the opponent. It was recently shown that both K-learning and UCB variants enjoy sub-linear regret bounds in this context [27]. Here we prove a regret bound for policies in the optimistic set under a sub-Gaussian assumption.

**Theorem 3.** *Let* alg *produce any sequence of policies* $\pi^t$, $t = 1, \ldots, T$, *that satisfy* $\pi^t \in \mathcal{P}_{\phi,\Lambda}^t$ *and let the opponent produce any policies* $\lambda^t$, $t = 1, \ldots, T$, *adapted to* $\mathcal{F}_t$. *Assuming that the prior over each entry of* $R$ *and reward noise are* $1$-*sub-Gaussian we have*

$$\mathrm{BayesRegret}(\phi, \mathrm{alg}, T) \leq \sqrt{2AmT \log A(1 + \log T)} = \tilde{O}(\sqrt{mAT}).$$

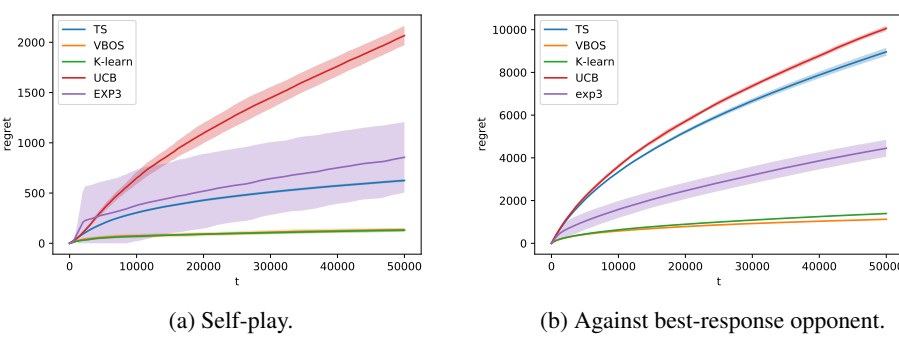

(a) Self-play.  (b) Against best-response opponent.

Figure 2: Regret on $50 \times 50$ random game.

In Figure 2 we show how five agents perform on a $50 \times 50$ randomly generated game in self-play and against a best-response opponent. The entries of $R$ were sampled from prior $\mathcal{N}(0, 1)$ and the noise term $\eta^t$ at each time-period was also sampled from $\mathcal{N}(0, 1)$. In self-play the opponent has the same knowledge and is applying the same algorithm to generate the policy at each time-step. The best-response opponent knows the matrix $M$ exactly as well as the policy that the agent produces and picks the action that minimizes the reward to the agent. The matrix is the same in both cases. We compare the K-learning and UCB algorithms as described in [27] as well as EXP3 [3], a generic adversarial bandit algorithm, VBOS Algorithm 4, and TS Algorithm 3 over 8 seeds. As can be seen VBOS is the best performing algorithm in both cases, followed closely by K-learning. Although performing moderately well in self-play, it is no surprise that TS does very poorly against the best-response opponent since it is sometimes susceptible to exploitation, as we show next.

Consider the two-player zero-sum game counter-example presented in [27]. Here $\Lambda = \Delta_2$, and

$$R = \begin{bmatrix} r & 0 \\ 0 & -1 \end{bmatrix}, \quad r = \begin{cases} 1 & \text{with probability } 1/2 \\ -1 & \text{with probability } 1/2. \end{cases} \quad (12)$$

In this case TS will play one of two policies: $\pi^{\mathrm{TS},1} = [1, 0]$ or $\pi^{\mathrm{TS},2} = [1/2, 1/2]$, each with probability $1/2$. A quick calculation yields $\mathbb{E}V_{R,\Lambda}^\star = -1/4$ and $\mathcal{G}_{\phi,\Lambda}(\pi^{\mathrm{TS},2}) = -1/2$, so with probability $1/2$ TS will produce a policy *not* in the optimistic set, since $\mathcal{G}_{\phi,\Lambda}(\pi^{\mathrm{TS},2}) < \mathbb{E}V_{R,\Lambda}^\star$. Now, consider the case where $r = 1$ and the opponent plays the strategy $[0, 1]$ in every round. Then every time TS produces $\pi^{\mathrm{TS},2}$ it will incur expected regret of $1/4$. Since the value of $r$ is never revealed the uncertainty is never resolved and so this pattern continues forever. This manifests as linear regret for TS in this instance. By contrast VBOS always produces $\pi^{\mathrm{VBOS}} = [1, 0]$ and so suffers no regret.

In Figure 3 we show a realization of $\mathcal{P}_{\phi,\Lambda}^t$ over time for an agent following the VBOS policy in a randomly generated $3 \times 3$ zero-sum two-player game. Observe that the TS algorithm has produced policies outside the optimistic set, but VBOS remains inside at all times. Note that this does not *prove* that TS will suffer linear regret, since membership of the optimistic set is only a sufficient, not necessary, condition.

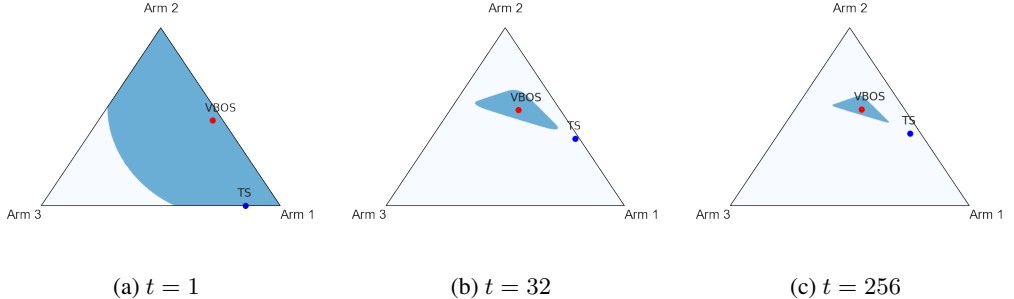

(a) $t = 1$        (b) $t = 32$        (c) $t = 256$

Figure 3: Policy simplex for a bilinear saddle-point problem. The shaded area is the optimistic set $\mathcal{P}^t_{\phi,\Lambda}$. The blue dot is the policy generated by TS and the red dot is the policy generated by VBOS. Notice that TS has produced policies outside of the optimistic set.

## 4.2 Constrained bandits

In constrained bandits we want to find the policy that maximizes expected reward subject to a set of constraints. The constraints specify that the expected value of other reward functions associated with the same arms, which are also initially unknown, are larger than zero. Formally, we have the usual unknown reward vector $\mu_1 \in \mathbb{R}^A$ we want to maximize as well as $m-1$ other unknown reward vectors $\mu_2, \ldots, \mu_m \in \mathbb{R}^A$ that define the constraints. When there is zero uncertainty we want to solve the following primal-dual linear-programming pair

$$
\begin{array}{ll}
\text{maximize} & \mu_1^\top \pi \\
\text{subject to} & \mu_i^\top \pi \geq 0, \quad i = 2, \ldots, m \\
& \pi \in \Delta_A
\end{array}
\qquad\qquad
\begin{array}{ll}
\text{minimize} & V \\
\text{subject to} & \sum_{i=1}^m \mu_i \lambda_i \leq V\mathbf{1} \\
& \lambda \geq 0, \lambda_1 = 1,
\end{array}
$$

over primal variable $\pi \in \Delta_A$ and dual variables $V \in \mathbb{R}$ and $\lambda \in \mathbb{R}^m$, and where $\mathbf{1}$ denotes the vector of all ones. We denote by $(\pi^\star, \lambda^\star)$ any optimal primal-dual points.

When there is uncertainty in the reward vectors, then we must be careful to ensure that the Bayesian regret is well-defined. If the support of the prior includes infeasible problems, which have optimal value $-\infty$, then the expectation of the regret under that prior is unbounded below. However, if we make the assumption that $\|\lambda^\star\|_2 \leq C$ for some $C > 0$ $\phi$-almost surely, which implies that the problem is feasible almost surely, then the Bayesian regret is well-defined and we can derive a regret bound that will depend on $C$. With this in mind we define the set $\Lambda = \{\lambda \in \mathbb{R}^m \mid \lambda_1 = 1, \lambda \geq 0, \|\lambda\|_2 \leq C\}$. Let $R \in \mathbb{R}^{m \times A}$ be the matrix with the reward vectors $\mu_1, \ldots, \mu_m$ stacked row-wise, then we can write the Lagrangian for the above primal-dual pair as Equation (7), *i.e.*, this problem is a constrained bilinear saddle-point problem in terms of the primal-dual variables $\pi$ and $\lambda$.

At each round $t$ the agent produces a policy $\pi^t$ from which a column is sampled and the agent receives observation vector $o_t = (\mu_{1j} + \eta^t_{1j}, \mu_{2j} + \eta^t_{2j}, \ldots, \mu_{mj} + \eta^t_{mj})$ where $j$ is the index of the sampled column and $\eta^t_{ij}$ is zero-mean noise for entry $i, j$. We define the reward at each time-period to be $r_t = \min_{\lambda \in \Lambda} \lambda^\top R \pi^t$. We choose this reward for two reasons, firstly the regret with this reward is always non-negative, and secondly if the regret is zero then the corresponding policy $\pi^t$ is optimal for the linear program since it is a maximizer of the saddle-point problem [4]. Unlike the previous examples the agent never observes this reward since it depends on exact knowledge of $R$ to compute, the reward is only used to measure regret. Nevertheless, we have the following Bayesian regret bound for policies in the optimistic set.

**Theorem 4.** *Let* alg *produce any sequence of policies* $\pi^t$, $t = 1, \ldots, T$, *that satisfy* $\pi^t \in \mathcal{P}^t_{\phi,\Lambda}$ *and assume that the prior over each entry of* $R$ *and reward noise are 1-sub-Gaussian and that* $\|\lambda^\star\|_2 \leq C$ $\phi$*-almost surely, then*

$$
\text{BayesRegret}(\phi, \text{alg}, T) \leq C\left(\sqrt{2 \log A(1 + \log T)} + 2\sqrt{m}\right)\sqrt{AT} = \tilde{O}(\sqrt{mAT}).
$$

Next we present a numerical experiment with $n = 50$ variables and $m = 25$ constraints. The entries of $R$ were sampled from prior $\mathcal{N}(-0.15, 1)$, and we repeatedly sampled from this distribution until

a feasible problem was found. We chose a slightly negative bias in the prior for $R$ in order to make the problem of finding a feasible policy more challenging. All noise was sampled from $\mathcal{N}(0, 1)$. For this experiment we set $C = 10$; the true optimal dual variable satisfied $\|\lambda^\star\| \approx 7$. In Figure 4 we compare the performance of TS Algorithm 3, VBOS Algorithm 4, and K-learning [23] over 8 seeds. We also plot how much the policy produced by each algorithm violated the constraints at each time-period, as measured by $\ell_\infty$ norm. As can be seen both VBOS and K-learning enjoy sub-linear regret and have their constraint violation converging to zero, with VBOS slightly outperforming K-learning on both metrics. TS on the other hand appears to suffer from linear regret, and the constraint violation decreases up to a point after which it saturates at a non-zero value. It is clear that TS is unable to adequately trade off exploration and exploitation in this problem.

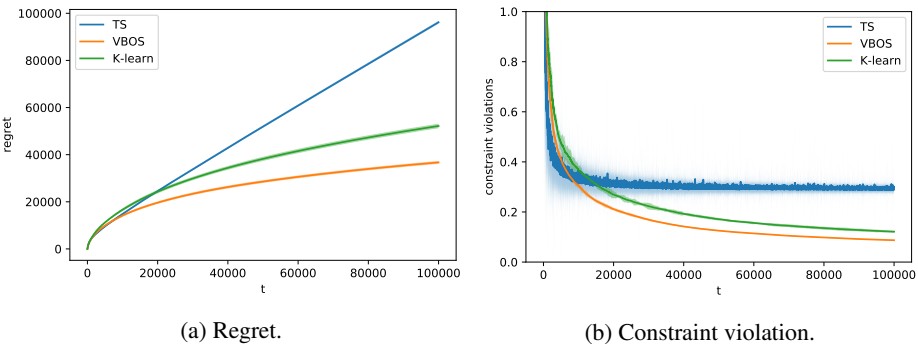

(a) Regret.            (b) Constraint violation.

Figure 4: Performance on a random $50 \times 25$ constrained bandit problem.

## 5   Conclusion

In this manuscript we introduced variational Bayesian optimistic sampling (VBOS), which is an algorithm that produces policies by solving an optimization problem in each round of a sequential decision problem. The optimization problem that must be solved is always convex, no matter how complicated the posteriors are. We showed that VBOS enjoys low regret for several problem families, including the well-known multi-armed bandit problem as well as more exotic problems like constrained bandits where TS can fail.

We conclude with some discussion about future directions for this work. First, in follow-up work we intend to show that VBOS is applicable in structured and multi-period problems like Markov decision processes, and again prove Bayesian regret guarantees. Second, an avenue we did not explore in this work was to parameterize the policy to be in a certain family, *e.g.*, linear in some basis. The variational formulation naturally extends to this case, where the optimization problem is in terms of the basis coefficients (which maintains convexity in the linear case). Combining these two suggests using the variational formulation to find the linearly parameterized VBOS policy in an MDP, which amounts to solving a convex optimization problem that may be much smaller than solving the corresponding problem without the parameterization. We expect that a regret bound is possible in this case under certain assumptions on the basis. Finally, since the VBOS objective is *differentiable* this opens the door to using differentiable computing architectures, such as deep neural networks trained using backpropagation. We speculate that it may be possible to apply VBOS in deep reinforcement learning to tackle the challenging exploration-exploitation tradeoff in those domains.

## Acknowledgments and Disclosure of Funding

The authors would like to thank Ben Van Roy, Satinder Singh, and Zheng Wen for discussions and feedback. The authors received no specific funding for this work.

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
