# A Appendix

**Lemma 1.** *Let $X : \Omega \to \mathbb{R}$ be a random variable on $(\Omega, \mathcal{F}, \mathbb{P})$ satisfying $X \in L_1$, and let $A \in \mathcal{F}$ be an event with $\mathbb{P}(A) > 0$. Then, for any $\tau \geq 0$*

$$\mathbb{E}[X|A] \leq \mathbb{E}X + \tau\Psi_X(1/\tau) - \tau \log \mathbb{P}(A). \tag{3}$$

*Proof.* Starting with the definition of conditional expectation, for any $\tau \geq 0$ we have

$$\begin{aligned}
\mathbb{E}[X|A] &= \frac{1}{\mathbb{P}(A)} \int_A X(\omega) d\mathbb{P}(\omega) \\
&= \frac{\tau}{\mathbb{P}(A)} \int_A \log \exp(X(\omega)/\tau) d\mathbb{P}(\omega) \\
&\leq \tau \log \left( \frac{1}{\mathbb{P}(A)} \int_A \exp(X(\omega)/\tau) d\mathbb{P}(\omega) \right) \\
&= \tau \log \int_A \exp(X(\omega)/\tau) d\mathbb{P}(\omega) - \tau \log \mathbb{P}(A) \\
&\leq \tau \log \int_\Omega \exp(X(\omega)/\tau) d\mathbb{P}(\omega) - \tau \log \mathbb{P}(A) \\
&= \mathbb{E}X + \tau\Psi_X(1/\tau) - \tau \log \mathbb{P}(A),
\end{aligned}$$

where we used Jensen's inequality in the third line. $\qquad\square$

**Theorem 1.** *Let $X : \Omega \to \mathbb{R}$ be a random variable such that the interior of the domain of $\Psi_X$ is non-empty, then under the same assumptions as Lemma 1 we have* [2]

$$\mathbb{E}[X|A] \leq \mathbb{E}X + (\Psi_X^*)^{-1}(-\log \mathbb{P}(A)).$$

*Proof.* This is a combination of Lemma 1 and Lemma 5 (presented next). Lemma 5 applies because $\Psi_X$ is Legendre type on $\mathbb{R}_+$ since the interior of the domain of $\Psi_X$ is non-empty [15, Thm. 2.3], and we have $\Psi_X(0) = 0$ and $\Psi_X'(0) = 0$. $\qquad\square$

**Lemma 5.** *Let $f : \mathbb{R}_+ \to \mathbb{R}$ be Legendre type with $f(0) = 0$, $f'(0) = 0$ and denote the convex conjugate of $f$ as $f^* : \mathbb{R} \to \mathbb{R}$, i.e.,*

$$f^*(p) = \sup_{x \geq 0} (xp - f(x)).$$

*Then for $y \geq 0$*

$$\inf_{\tau \geq 0} (\tau f(1/\tau) + \tau y) = (f^*)^{-1}(y).$$

*Proof.* The $y = 0$ case is straightforward, so consider the case where $y > 0$. First, $f^*$ is Legendre type by the assumption that $f$ is Legendre type (loosely, differentiable and strictly convex on the interior of its domain) [19, §26.4]. Since $f(0) = 0$ and $f'(0) = 0$ we have that $f^*(0) = 0$ and $(f^*)'(0) = 0$. These imply that $f^*$ is strictly increasing and continuous on $\mathbf{dom}\, f^* \cap \mathbb{R}_+$ and has range $\mathbb{R}_+$, so the quantity $(f^*)^{-1}(y)$ is well-defined for any $y \geq 0$ and moreover $(f^*)^{-1}(y) \geq 0$. The Fenchel-Young inequality states that

$$f(1/\tau) + f^*(p) \geq p/\tau,$$

for any $p, \tau \in \mathbb{R}_+$ with equality if and only if $(1/\tau) = (f^*)'(p)$. Fix $p = (f^*)^{-1}(y)$ and note that $p > 0$ since $y > 0$. Then for $\tau \geq 0$ we have

$$\tau f(1/\tau) + \tau y \geq (f^*)^{-1}(y).$$

Equality is attained by $\tau = 1/(f^*)'(p)$ with $0 \leq \tau < \infty$, since $y > 0$, $p \in \mathbf{dom}\, f^* \cap \mathbb{R}_+$, and $f^*$ is strictly increasing on $\mathbf{dom}\, f^* \cap \mathbb{R}_+$.

$\qquad\square$

---

[2] If $\Psi_X = 0$, then we take $(\Psi_X^*)^{-1} = 0$.

**Lemma 2.** *Let* $\mu : \Omega \to \mathbb{R}^A$, $\mu \in L_1^A$, *be a random variable, let* $i^\star = \arg\max_i \mu_i$ *(ties broken arbitrarily) and denote by* $\Psi_i := \Psi_{\mu_i}$, *then*

$$\mathbb{E} \max_i \mu_i \leq \sum_{i=1}^A \mathbb{P}(i^\star = i) \left( \mathbb{E}\mu_i + (\Psi_i^*)^{-1}(-\log \mathbb{P}(i^\star = i)) \right).$$

*Proof.* This follows directly from the definition of the maximum and Theorem 1.

$$\mathbb{E} \max_i \mu_i = \sum_{i=1}^A \mathbb{P}(i^\star = i)\mathbb{E}[\mu_i | i^\star = i] \leq \sum_{i=1}^A \mathbb{P}(i^\star = i) \left( \mathbb{E}\mu_i + (\Psi_i^*)^{-1}(-\log \mathbb{P}(i^\star = i)) \right).$$

$\square$

**Lemma 3.** *Let* alg *produce any sequence of policies* $\pi^t$, $t = 1, \ldots, T$, *that satisfy* $\pi^t \in \mathcal{P}_\phi^t$, *then*

$$\text{BayesRegret}(\phi, \text{alg}, T) \leq \mathbb{E} \sum_{t=1}^T \sum_{i=1}^A \pi_i^t (\Psi_i^{t*})^{-1}(-\log \pi_i^t).$$

*Proof.* Starting with the definition of Bayesian regret in Equation (2),

$$\begin{aligned}
\text{BayesRegret}(\phi, \text{alg}, T) &= \mathbb{E} \sum_{t=1}^T \left( \max_i \mu_i - r_t \right) \\
&= \mathbb{E} \sum_{t=1}^T \left( \mathbb{E}^t \max_i \mu_i - \sum_{i=1}^A \pi_i^t \mathbb{E}^t \mu_i \right) \\
&\leq \mathbb{E} \sum_{t=1}^T \sum_{i=1}^A \pi_i^t (\Psi_i^{t*})^{-1}(-\log \pi_i^t)
\end{aligned}$$

which follows from the tower property of conditional expectation and lemma 2. $\square$

**Theorem 2.** *Let* alg *produce any sequence of policies* $\pi^t$, $t = 1, \ldots, T$, *that satisfy* $\pi^t \in \mathcal{P}_\phi^t$ *and assume that both the prior and reward noise are* 1-*sub-Gaussian for each arm, then*

$$\text{BayesRegret}(\phi, \text{alg}, T) \leq \sqrt{2AT \log A(1 + \log T)} = \tilde{O}(\sqrt{AT}).$$

*Proof.* Since the prior and noise terms are 1-sub-Gaussian for each arm, we can bound the cumulant generating function of $\mu_i$ at time $t$ as

$$\Psi_i^t(\beta) \leq \frac{\beta^2}{2(n_i^t + 1)}, \tag{13}$$

where $n_i^t$ is the number of observations of arm $i$ before time $t$. A quick calculation yields the following bound for $y \geq 0$

$$(\Psi_i^{t*})^{-1}(y) \leq \sqrt{\frac{2y}{n_i^t + 1}}. \tag{14}$$

Combining this with Lemma (3),

$$\begin{aligned}
\text{BayesRegret}(\phi, \text{alg}, T) &\leq \mathbb{E} \sum_{t=1}^T \sum_{i=1}^A \pi_i^t (\Psi_i^{t*})^{-1}(-\log \pi_i^t) \\
&\leq \mathbb{E} \sum_{t=1}^T \sum_{i=1}^A \pi_i^t \sqrt{\frac{-2 \log \pi_t}{n_i^t + 1}} \\
&\leq \mathbb{E} \sqrt{\sum_{t=1}^T H(\pi^t) \sum_{t=1}^T \sum_{i=1}^A \frac{2\pi_t}{n_i^t + 1}}
\end{aligned}$$

which follows from Cauchy-Scwarz. To conclude the proof we use the fact that $H(\pi_t) \leq \log(A)$ and a pigeonhole principle included as Lemma 6. $\square$

**Lemma 6.** *Consider a process that at each time $t$ selects a single index $a_t$ from $\{1, \ldots, q\}$ with probability $p_{a_t}^t$. Let $n_i^t$ denote the count of the number of times index $i$ has been selected before time $t$. Then*

$$\mathbb{E} \sum_{t=1}^{T} \sum_{i=1}^{q} \frac{p_i^t}{n_i^t + 1} \leq q(1 + \log T).$$

*Proof.* This follows from an application of the pigeonhole principle,

$$\mathbb{E} \sum_{t=1}^{T} \sum_{i=1}^{q} \frac{p_i^t}{n_i^t + 1} = \mathbb{E} \sum_{t=1}^{T} \mathbb{E}_{a_t \sim p^t} (n_{a_t}^t + 1)^{-1}$$

$$= \mathbb{E}_{a_1 \sim p^1, \ldots, a_T \sim p^T} \mathbb{E} \sum_{t=1}^{T} (n_{a_t}^t + 1)^{-1}$$

$$= \mathbb{E}_{n_1^{T+1}, \ldots, n_q^{T+1}} \mathbb{E} \sum_{i=1}^{q} \sum_{t=1}^{n_i^{T+1}} 1/t$$

$$\leq q \sum_{t=1}^{T} 1/t$$

$$\leq q(1 + \log T).$$

$\square$

**Lemma 4.** *Assuming that $R \in L_1^{m \times A}$ we have*

$$\mathbb{E}^t V_{R,\Lambda}^\star \leq \max_{\pi \in \Delta_A} \mathcal{G}_{\phi,\Lambda}^t(\pi). \tag{10}$$

*Proof.* Using Jensen's inequality and then the upper bound in Equation 5 we obtain

$$\mathbb{E}^t V_{R,\Lambda}^\star = \mathbb{E}^t \min_{\lambda \in \Lambda} \max_{\pi \in \Delta_A} \lambda^\top R \pi$$

$$\leq \min_{\lambda \in \Lambda} \mathbb{E}^t \max_{\pi \in \Delta_A} \lambda^\top R \pi$$

$$= \min_{\lambda \in \Lambda} \mathbb{E}^t \max_j (R^\top \lambda)_j$$

$$\leq \min_{\lambda \in \Lambda, \tau \geq 0} \max_{\pi \in \Delta_A} \sum_{j=1}^{A} \pi_j (\lambda^\top \mathbb{E}^t R_j + \tau_j \Psi_j^t(\lambda/\tau_j) - \tau_j \log \pi_j) \tag{15}$$

$$= \max_{\pi \in \Delta_A} \min_{\lambda \in \Lambda, \tau \geq 0} \sum_{j=1}^{A} \pi_j (\lambda^\top \mathbb{E}^t R_j + \tau_j \Psi_j^t(\lambda/\tau_j) - \tau_j \log \pi_j)$$

$$= \max_{\pi \in \Delta_A} \mathcal{G}_{\phi,\Lambda}(\pi),$$

where we could swap the min and max using the minimax theorem. $\square$

**Theorem 3.** *Let* alg *produce any sequence of policies $\pi^t$, $t = 1, \ldots, T$, that satisfy $\pi^t \in \mathcal{P}_{\phi,\Lambda}^t$ and let the opponent produce any policies $\lambda^t$, $t = 1, \ldots, T$, adapted to $\mathcal{F}_t$. Assuming that the prior over each entry of $R$ and reward noise are 1-sub-Gaussian we have*

$$\mathrm{BayesRegret}(\phi, \mathrm{alg}, T) \leq \sqrt{2AmT \log A(1 + \log T)} = \tilde{O}(\sqrt{mAT}).$$

*Proof.* This is a straightforward extension of the techniques in Theorem 2. First, let us denote by

$$\mathcal{L}_R^t(\pi, \lambda, \tau) = \sum_{j=1}^{A} \pi_j (\lambda^\top \mathbb{E}^t R_j + \tau_j \Psi_j^t(\lambda/\tau_j) - \tau_j \log \pi_j).$$

Since the prior and noise terms are 1-sub-Gaussian for each entry of $R$, we can bound the cumulant generating function of $R_j$ at time $t$ as

$$\Psi_j^t(\beta) \le \sum_{i=1}^m \frac{\beta_i^2}{2(n_{ij}^t + 1)}, \tag{16}$$

where $n_{ij}^t$ is the number of observations of entry $i, j$ before time $t$. Then, using Lemma 4

$$
\begin{aligned}
\text{BayesRegret}(\phi, \text{alg}, T) &= \mathbb{E} \sum_{t=1}^T (V_{R,\Lambda}^\star - r_t) \\
&= \mathbb{E} \sum_{t=1}^T \mathbb{E}^t (V_{R,\Lambda}^\star - r_t) \\
&\le \mathbb{E} \sum_{t=1}^T \left( \min_{\lambda \in \Lambda, \tau \ge 0} \mathcal{L}_R^t(\pi^t, \lambda, \tau) - (\lambda^t)^\top (\mathbb{E}^t R)\pi^t \right) \\
&\le \mathbb{E} \sum_{t=1}^T \left( \min_\tau \mathcal{L}_R^t(\pi^t, \lambda^t, \tau) - (\lambda^t)^\top (\mathbb{E}^t R)\pi^t \right) \\
&= \sum_{t=1}^T \sum_{j=1}^A \pi_j^t \min_{\tau_j} (\tau_j \Psi_j^t(\lambda^t/\tau_j) - \tau_j \log \pi_j^t) \\
&\le \sum_{t=1}^T \sum_{j=1}^A \pi_j^t \min_{\tau_j} \left( \sum_{i=1}^m \frac{(\lambda_i^t)^2}{2\tau_j(n_{ij}^t + 1)} - \tau_j \log \pi_j^t \right) \\
&= \sum_{t=1}^T \sum_{j=1}^A \pi_j^t \sqrt{ \sum_{i=1}^m \frac{-2(\lambda_i^t)^2 \log \pi_j^t}{(n_{ij}^t + 1)} } \\
&\le \sqrt{ \sum_{t=1}^T H(\pi^t) } \sqrt{ 2 \sum_{t=1}^T \sum_{i,j} \frac{\lambda_i^t \pi_j^t}{(n_{ij}^t + 1)} } \\
&\le \sqrt{2mAT(\log A)(1 + \log T)},
\end{aligned}
$$

where we used the sub-Gaussian bound Eq. (16), Cauchy-Schwarz, the fact that $\lambda^t$ is a probability distribution adapted to $\mathcal{F}_t$, and the pigeonhole principle Lemma 6. $\qquad\square$

**Theorem 4.** *Let* alg *produce any sequence of policies* $\pi^t$, $t = 1, \ldots, T$, *that satisfy* $\pi^t \in \mathcal{P}_{\phi,\Lambda}^t$ *and assume that the prior over each entry of $R$ and reward noise are 1-sub-Gaussian and that* $\|\lambda^\star\|_2 \le C$ $\phi$-almost surely, then

$$\text{BayesRegret}(\phi, \text{alg}, T) \le C \left( \sqrt{2 \log A(1 + \log T)} + 2\sqrt{m} \right) \sqrt{AT} = \tilde{O}(\sqrt{mAT}).$$

*Proof.* Let $\lambda^{t\star} = \arg\min_{\lambda \in \Lambda} \lambda^\top R \pi_t$, which exists for any fixed $R$ since the set $\Lambda$ is compact and the objective function is linear. Note that $R$ is a random variable, and so $\lambda^{t\star}$ is also a random variable for all $t$, and note that the reward we defined at time $t$ is given by $r_t = (\lambda^{t\star})^\top R \pi_t$. Let us denote by

$$\mathcal{L}_R^t(\pi, \lambda, \tau) = \sum_{j=1}^A \pi_j (\lambda^\top \mathbb{E}^t R_j + \tau_j \Psi_j^t(\lambda/\tau_j) - \tau_j \log \pi_j).$$

Since the prior and noise terms are 1-sub-Gaussian for each entry of $R$, we can bound the cumulant generating function of $R_j$ at time $t$ as

$$\Psi_j^t(\beta) \le \frac{\|\beta\|^2}{2(n_j^t + 1)}, \tag{17}$$

where $n_j^t$ is the number of observations of column $j$ before time $t$ (the agent observes outcomes from all rows, hence the use of $n_j^t$ rather than $n_{ij}^t$). Now with the definition of the Bayesian regret and using Lemma 4

$$
\begin{aligned}
\text{BayesRegret}(\phi, \text{alg}, T, \mu) &= \mathbb{E} \sum_{t=1}^{T} (V_{R,\Lambda}^\star - r_t) \\
&\leq \mathbb{E} \sum_{t=1}^{T} \left( \min_{\lambda \in \Lambda, \tau \geq 0} \mathcal{L}_R^t(\pi^t, \lambda, \tau) - \mathbb{E}^t((\lambda^{t\star})^\top R \pi_t) \right) \\
&\leq \mathbb{E} \sum_{t=1}^{T} \left( \min_\tau \mathcal{L}_R^t(\pi^t, \mathbb{E}^t \lambda^{t\star}, \tau) - \mathbb{E}^t((\lambda^{t\star})^\top R \pi^t) \right).
\end{aligned}
$$

Now we write the last line above as

$$
\mathbb{E} \sum_{t=1}^{T} \left( \min_\tau \mathcal{L}(\pi^t, \mathbb{E}^t \lambda^{t\star}, \tau) - (\pi^t)^\top \mathbb{E}^t R \mathbb{E}^t \lambda^{t\star} + (\pi^t)^\top \mathbb{E}^t R \mathbb{E}^t \lambda^{t\star} - (\pi^t)^\top \mathbb{E}^t (R \lambda^{t\star}) \right),
$$

which we shall bound in two parts. First, we use the standard approach we have used throughout this manuscript. Using Eq. (17).

$$
\begin{aligned}
\mathbb{E} \sum_{t=1}^{T} \left( \min_\tau \mathcal{L}(\pi^t, \mathbb{E}^t \lambda^{t\star}, \tau) - (\pi^t)^\top \mathbb{E}^t R \mathbb{E}^t \lambda^{t\star} \right) &= \mathbb{E} \sum_{t=1}^{T} \sum_{i=1}^{A} \pi_i^t \min_{\tau_i \geq 0} \left( \tau_i \Psi_i^t (\mathbb{E}^t \lambda^{t\star} / \tau_i) - \tau_i \log \pi_i^t \right) \\
&\leq \mathbb{E} \sum_{t=1}^{T} \sum_{i=1}^{A} \pi_i^t \min_{\tau_i \geq 0} \left( \frac{\|\mathbb{E}^t \lambda^{t\star}\|_2^2}{2\tau_i(n_i^t + 1)} - \tau_i \log \pi_i^t \right) \\
&= \mathbb{E} \sum_{t=1}^{T} \sum_{i=1}^{A} \pi_i^t \sqrt{\frac{2(-\log \pi_i^t)\|\mathbb{E}^t \lambda^{t\star}\|_2^2}{n_i^t + 1}} \\
&\leq C \mathbb{E} \sqrt{2 \left( \sum_{t=1}^{T} H(\pi^t) \right) \left( \sum_{t=1}^{T} \sum_{i=1}^{A} \frac{\pi_i^t}{n_i^t + 1} \right)} \\
&\leq C \sqrt{2TA \log A (1 + \log T)},
\end{aligned}
$$

where we used the sub-Gaussian property Eq. (17), Cauchy-Schwarz, the fact that $H(\pi^t) \leq \log A$, and the fact that $\|\lambda^{t\star}\| \leq C$ almost surely which implies that $\|\mathbb{E}^t \lambda^{t\star}\| \leq C$, due to Jensen's inequality.

Before we bound the remaining term observe that if zero-mean random variable $X : \Omega \to \mathbb{R}$ is $\sigma$-sub-Gaussian, then the variance of $X$ satisfies $\mathbf{var} X \leq \sigma^2$, which is easily verified by a Taylor expansion of the cumulant generating function. Since the prior and noise terms are 1-sub-Gaussian for each entry of $R$ this implies that

$$
\mathbf{var}^t R_{ij} \leq (n_i^t + 1)^{-1}, \tag{18}
$$

where $\mathbf{var}^t$ is the variance conditioned on $\mathcal{F}_t$ and as before $n_i^t$ is the number of times column $i$ has been selected by the agent before time $t$. We bound the remaining term as follows

$$
\mathbb{E} \sum_{t=1}^{T} \left( (\pi^t)^\top \mathbb{E}^t R \mathbb{E}^t \lambda^{t\star} - (\pi^t)^\top \mathbb{E}^t(R\lambda^{t\star}) \right) = \mathbb{E} \sum_{t=1}^{T} (\pi^t)^\top \mathbb{E}^t((\mathbb{E}^t R - R)\lambda^{t\star})
$$

$$
= \mathbb{E} \sum_{t=1}^{T} \sum_{i=1}^{A} \pi_i^t \mathbb{E}^t((\mathbb{E}^t R_i - R_i)^\top \lambda^{t\star})
$$

$$
\leq \mathbb{E} \sum_{t=1}^{T} \sum_{i=1}^{A} \pi_i^t \sqrt{\mathbb{E}^t \|R_i - \mathbb{E}^t R_i\|_2^2 \mathbb{E}^t \|\lambda^{t\star}\|_2^2}
$$

$$
\leq C\mathbb{E} \sum_{t=1}^{T} \sum_{i=1}^{A} \pi_i^t \sqrt{\sum_{j=1}^{m} \mathbf{var}^t R_{ij}}
$$

$$
\leq C\mathbb{E} \sum_{t=1}^{T} \sum_{i=1}^{A} \pi_i^t \sqrt{m/(n_i^t + 1)}
$$

$$
\leq 2C\sqrt{TAm},
$$

where we used Cauchy-Schwarz, the fact that $\|\lambda^{t\star}\| \leq C$ almost surely, the sub-Gaussian bound on the variance of $R_{ij}$ from Eq. (18), and a pigeonhole principle. Combining the two upper bounds we have

$$
\text{BayesRegret}(\phi, \text{alg}, T) \leq C\left( \sqrt{2\log A(1 + \log T)} + 2\sqrt{m} \right)\sqrt{AT}.
$$

$\square$

## B  Compute requirements

All experiments were run on a single 2017 MacBook Pro.