# OpenReview forum: "Variational Bayesian Optimistic Sampling"
_NeurIPS.cc/2021/Conference — NeurIPS 2021 Spotlight_

### Official Review · Reviewer_zmC9 · 2021-07-16

**Rating:** 7
**Confidence:** 2

**Summary:**

This paper presents and establishes regret bounds for a new heuristic, inspired by Thompson sampling, for selecting actions in stochastic multi-arm bandit and bilinear saddle-point problems. As in Thompson sampling, a belief state is maintained over the values of each arm. However, instead of selecting an action according to its probability of optimality, one instead computes a policy (distribution over actions) that maximizes a novel objective. The objective can be understood as combining a "pure exploitation" term (the expected reward of the policy, under the current belief distribution), and an "exploration" term that depends only on the relative concentrations (and not the expected values) of the posteriors for each arm.

**Ethical Concerns:**

None that I can see.

**Limitations And Societal Impact:**

Limitations of the proposed method are not clearly discussed, as far as I can see. I would be interested to see a more complete discussion of the conditions under which VTS can be applied (i.e., the conditions under which the VTS objective, or its derivative / maximum, can be efficiently computed or estimated).

**Main Review:**

Overall, this seems like a very nice paper. I appreciated the clear logical progression, from the definition of the stochastically optimistic set, to the presentation of VTS, to the bilinear saddle-point problem extension. I also appreciated the intuition provided by Section 3 and by the examples depicted in the figures. The motivations for VTS (applicability to games while maintaining sub-linear regret, differentiability) seem plausible to me, and although this paper does not present especially compelling empirical evidence of VTS's applicability in practice, it is a nice, self-contained theory paper. I do have a couple questions / concerns:

1. One motivation offered in the introduction to look beyond TS is that it may require samples from intractable posteriors over the values for each arm. Could you say a bit more about when VTS is tractable, but sampling from these posteriors is not? It seems you do need to know or estimate the expectation of each posterior (for the "exploit" term in the objective), and you also need to compute convex conjugates of inverse cumulant generating functions. Is this really "far easier than sampling [in many cases]" (L41)? What cases?

2. The name "Variational Thompson Sampling" is misleading, as you acknowledge in L142. (To my ear, the name suggests an algorithm that uses variational inference to approximate the per-arm posterior distributions.) At multiple points in the paper you describe the VTS distribution as a "variational approximation" (in Fig. 1, an approximation "to the posterior probability of optimality"), but is there any sense in which it is a variational approximation to that distribution? It doesn't seem like it, other than that in this particular example bandit problem, the distributions happen to be somewhat close. The connection to the ELBO (L188) also seemed very tenuous, though perhaps I'm missing something.

3. In Section 3, you write that K-learning's Eq (5) is equivalent to VTS. In your experiments, what accounts for the minor discrepancy between K-learning and VTS? Do you let the temperature parameter vary by arm?

Despite these questions, this was an interesting paper that I enjoyed reading, and everything seems technically sound—so I am inclined to accept. I am, however, marking my confidence rather low (I think I've followed your presentation, but I feel less equipped to evaluate originality (particularly e.g. w.r.t K-learning) or significance).

Minor comments:

1. On line 61, should it be $\mu_{a_t}$ instead of $\mu_{i_t}$?
2. The notation $\eta_{a_t}$ suggests an action-dependent noise that does not vary with time (since the $t$ subscripts the $a$, not $\eta$). But presumably there is independent noise at each time step?

**Time Spent Reviewing:**

4.5 hours

---

> ### Author Response · Authors · 2021-08-09
> **Response to reviewer zmC9**
>
> Thank you for your time and consideration!
>
> * "One motivation offered in the introduction to look beyond TS is that it may require samples from intractable posteriors over the values for each arm. Could you say a bit more about when VTS is tractable, but sampling from these posteriors is not?"
>
> Thanks for raising this, we have clarified our discussion in the paper and de-emphasized this point somewhat so as to not make any extra broad claims that we do not concretely support. The basic idea is that in some cases sampling from an intractable posterior is very hard, but (approximately) integrating with respect to the pdf is easier. For example, if the prior and the likelihood are not conjugate, then the posterior is unlikely to have a simple form and sampling can be challenging. However, in these cases we can often use techniques like quadrature to integrate with respect to the pdf. In this case the expectation becomes a weighted sum, which is still convex in the parameter and therefore can be optimized.
>
> * "The name "Variational Thompson Sampling" is misleading”
>
>  Agreed, how do you feel about VOBS? The word 'variational' is used to mean that the policy is derived by solving an optimization problem, rather than, say, variational inference approximations.
>
> * “The connection to the ELBO (L188) also seemed very tenuous, though perhaps I'm missing something.”
>
> Agreed, that was tenuous, we have removed it.
>
> * "In Section 3, you write that K-learning's Eq (5) is equivalent to VTS. In your experiments, what accounts for the minor discrepancy between K-learning and VTS? Do you let the temperature parameter vary by arm?”
>
> K-learning is equivalent to VTS *only* when VTS is restricted to using the same temperature for each arm, which we did not do. VTS gets tighter upper bounds on the expected value of the max than K-learning due to this additional flexibility. We have re-run the experiments with more arms to show how this difference can affect performance in practice - as the number of arms increases the performance gap between VTS and K-learning increases too.
>
> * "On line 61, should it be μat instead of μit?"
>
> Yes, thank you, fixed.
>
> * "The notation ηat suggests an action-dependent noise that does not vary with time (since the t subscripts the a, not η). But presumably there is independent noise at each time step?”
>
> You’re right, we have added a $t$ superscript, since it depends on both the action and the time. Thanks!

---

> > ### Comment · Reviewer_zmC9 · 2021-08-20
> > **Thanks for the response!**
> >
> > Thank you for your clear response to these questions!
> >
> > I do think VOBS is a better name, and I can see the justification for calling the algorithm "variational." I also appreciate the clarification re: K-learning.
> >
> > On the first point, re: tractability, I am not totally convinced: when approximate integration is feasible (e.g. using quadrature), I believe approximate posterior sampling (e.g. via sampling-importance-resampling) shouldn't be much more difficult. One could try to make the argument that Thompson sampling is somehow "more hobbled" by approximate posterior samples than VOBS is by approximate integration—if this is the case, it would be interesting to see the justification spelled out (or to see the claim validated empirically in an example). At the least, if you plan to keep this claim in the paper, is there a reference you could cite on the relative hardness of integration and sampling, or e.g. on regimes where integration is easier than sampling?
> >
> > In any case, I still think this is a nice paper and continue to recommend acceptance. Thanks again for the response!

---

> > > ### Author Response · Authors · 2021-08-23
> > > **Thank you reviewer zmC9**
> > >
> > > Sorry for the delay in responding, I am traveling with spotty wifi access.
> > >
> > > Thanks again for your feedback. I think you're right, the claim as it stands is simply not supported by the rest of the paper. I think the correct thing would be to water down the claim, to something like "in vanilla bandits the VOBS approach offers an alternative to TS which may be more tractable in some cases". Does this sound reasonable?
> > >
> > > Personally, I think it should be possible to justify the claim (just as an aside here, I think watering down the claim is best for the paper). In Phan et al. "Thompson sampling and approximate inference" they showed that TS can suffer linear regret with even very small inference error, so TS is quite sensitive to errors that could arise in approximate sampling. We would need to show that under similar inference errors VOBS is more robust. I have some preliminary experimental results that demonstrate this - e.g., when the likelihood is misspecified TS tends to do worse than VOBS (depending on the degree of misspecification). Moreover, in work that we cut due to space constraints we derived another variational approximation that required no integration, only solving an optimization problem which was convex if the likelihood and prior terms were log-concave, and the resulting policy has the same regret bound in sub-Gaussian case. Lastly, when running something like MCMC or SIR to generate samples for TS, the same samples could be used to estimate the expectation required for VOBS. I will run an experiment to see how much better or worse VOBS is vs TS in this case - intuitively I would think that since VOBS is able to use all the samples to estimate the expectation (rather than just a single sample in SIR or the final one in the chain for MCMC) it should provide better estimates and have better performance (and typically we can control the error bound on an average of RVs better than we can for an MCMC chain). At this point this is just speculation though, so watering down the claim seems most appropriate.

---

### Official Review · Reviewer_TWH3 · 2021-07-16

**Rating:** 7
**Confidence:** 3

**Summary:**

This paper proposes a novel class of policies for online learning, named variational Thompson sampling (VTS). VTS is derived from a family of stochastically optimistic policies, which includes standard TS as a special case. VTS is amenable to some interesting interpretations when compared with difference existing policies such as UCB and K-learning, and is shown to outperform TS consistently in two classes of problems, i.e., zero-sum two player games and constrained bandits.

**Limitations And Societal Impact:**

Yes.

**Main Review:**

I find the derivation process of the VTS algorithm elegant and interesting. In particular, the paper starts from the definition of Bayesian regret and derives a related general inequality about conditional expectation, which naturally gives rise to a family of stochastically optimistic algorithms encompassing TS as a special case; VTS is then derived from this family of policies. Figure 1 and Figure 2 give nice illustrations to show that the VTS policy is closely related to yet different from TS. The connections of VTS to other existing policies in Section 3 also provide interesting insights. Section 4 gives two special cases where VTS has clear advantage over TS, supported by empirical simulations. Below are some questions/suggestions I have for the paper.

- I think the name "variational TS" may be misleading. Even after reading the abstract, I still get the feeling that the paper is in fact about proposing a variational inference algorithm to approximate standard TS. This has also been discussed in the paragraph from line 140 onwards, and I would suggest considering a less ambiguous name for the proposed method.
- Section 2.3: I wonder whether VTS is advantageous compared with standard TS for general multi-armed bandit problems (not considering the two special cases in Section 4)? It has been mentioned (lines 40-42) that solving for the VTS policy is easier than the sampling required by TS, but I think some illustrating examples would help a lot.
- Lines 214-222: the discussion here seems to suggest that VTS may actually be more related to K-learning than to TS. Moreover, in the simulation results in Section 4 (Figures 3 and 4), K-learning is very competitive with VTS. Therefore, I wonder will it be more natural for the paper to introduce VTS as a generalization of K-learning from the beginning, instead of deriving VTS in the way as in the current paper (Sections 2.1-2.3)?
- Line 263, Theorem 3: in this setting of zero-sum game between two players, does K-learning also have the same Bayesian regret as VTS?
- Lines 265-268: is there an intuitive explanation as to why VTS is not open to exploitation by the opponent?
- Figure 4: is there an explanation on the unsatisfactory performance of TS in this setting? Does the same explanation from section 4.1 (lines 228-230) also apply here?

**Time Spent Reviewing:**

6 hours

---

> ### Author Response · Authors · 2021-08-09
> **Response to reviewer TWH3**
>
> Thank you for your comments and suggestions!
>
> * “I think the name "variational TS" may be misleading.”
>
> We agree, do you think VOBS (Variational Bayesian Optimistic Sampling) is a better name?
>
> * “It has been mentioned (lines 40-42) that solving for the VTS policy is easier than the sampling required by TS, but I think some illustrating examples would help a lot.”
>
> Good point, we have rewritten that section to provide some better context and de-emphasized this point somewhat so as to not make any extra broad claims that we do not concretely support. The basic idea is that in certain cases it is easier to (approximately) integrate with respect to a pdf than to sample from that pdf, such as when the prior and the likelihood are not conjugate. In these cases we can often use techniques like quadrature to integrate with respect to the pdf. In which case the expectation becomes a weighted sum, which is still convex in the parameter and therefore can be optimized.
>
> * “VTS may actually be more related to K-learning than to TS”
>
> Unfortunately we don’t know of a way to derive VTS from K-learning, only to derive VTS separately then to show how it relates to K-learning after the fact (via the temperatures). Since we will rename the algorithm we can better make the connections in a more neutral fashion, ie, VOBS is closely related to (but distinct from) both K-learning and Thompson sampling. We re-ran the experiments to emphasize the performance benefits of VOBS/VTS over K-learning for larger problems.
>
> * “in this setting of zero-sum game between two players, does K-learning also have the same Bayesian regret as VTS?”
>
> Yes, K-learning has the same regret bound (up to log-factors) as VTS in these settings see “Matrix games with bandit feedback” O’Donoghue et al 2021. We have made this point more clearly in the text.
>
> * “ is there an intuitive explanation as to why VTS is not open to exploitation by the opponent?"
>
> We have expanded on this in the text. In short the difference is that VTS is guaranteed to be *optimistic* about every strategy (since it is constructing upper bounds on each entry). When it is optimistic, it is either correct or it is incorrect and to exploit it would require revealing information to it. On the other hand Thompson sampling is sometimes pessimistic, and in those cases it can be exploited without revealing information.
>
> * “is there an explanation on the unsatisfactory performance of TS in this setting? Does the same explanation from section 4.1 (lines 228-230) also apply here?”
>
> Good question, the basic idea is the same. TS can suffer large regret without learning anything about the problem. Note that we can rewrite the two-player zero-sum game as a particular constrained bandit problem (using duality) and so the counter-example that exists for the game case also acts as a counter-example for the constrained bandit case. We have added this to the manuscript.

---

> > ### Comment · Reviewer_TWH3 · 2021-08-21
> > **Quick Response**
> >
> > I appreciate the response from the authors, and the explanations all seem reasonable to me.
> >
> > Yes, I think "Variational Bayesian Optimistic Sampling" is an appropriate name, it should be able to avoid misunderstandings regarding its connection with TS.

---

### Official Review · Reviewer_AmXN · 2021-07-17

**Rating:** 6
**Confidence:** 4

**Summary:**

This work develops an analog to Thompson sampling by upper-bounding the expected regret in sequential decision-making problems. The two terms in the upper bound loosely resemble the evidence lower bound of variational inference: the first term encourages selecting arms with high expected reward; the second term depends on the inverse of the rate function and penalizes heavy tails and encourages exploration. The resulting variational Thompson sampling algorithm is evaluated on a random game and a constrained bandit problem.

**Limitations And Societal Impact:**

Yes

**Main Review:**

Originality:
- I read this work in the spirit of Wainwright and Jordan (2008; https://people.eecs.berkeley.edu/~wainwrig/Papers/WaiJor08_FTML.pdf) where variational inference is connected to marginal polytopes. While theoretical, I appreciate the connections the authors make between bounds dependent on inverse rate functions to concave sets of policy distributions, with Thompson sampling as a special case.

Quality:
- I did not evaluate proofs of the theorems; they seem sound at first glance. The empirical evaluation results in a favorable comparison to Thompson sampling.

Clarity:
- The work will benefit from significant revision in terms of clarity.
- Specific writing feedback:
- Split the first paragraph on Line 24; the paragraph on UCB on line 192 is too dense; several other 'walls of text'
- Put the noun/main character/subject of a sentence more front-and-center. I circled several things that made the writing confusing: "in this paper"; "at every step"; "this problem"; "the problem"; "the only requirement"; "in many cases"; "with that in mind"; "this simple strategy"; "have the laws of the posterior"; "with this notation"; "this brings us to"; "examining the problem that is solved"; "we conclude with some discussion"; "in this manuscript". The key point here is ctrl/cmd+F and searching for words like "it", "this", "that" and seeing if they can be replaced or the phrases can be cut. This lightens the cognitive load on a reader to figure out what "this" and other vague referents are, throughout the paper. Ditto for
- Consider giving $\mathcal{G}_\mu$ a name - it will make it easier on readers. For example, it may be the 'regret map', mapping natural parameters of the mean reward $\mu$ to their upper bound on regret. Wainwright and Jordan (2008) may give inspiration for names in terms of the marginal polytope, natural parameters, etc.
- Further, a visual aid of $\mathcal{G}_\mu$ would go a long way. It represents a concave set of probability distributions, and may be visualized using a diagram, perhaps as in Wainwright and Jordan (08).
- The abstract states "it permits the use of TS in a differential computation framework, such as a layer in a neural network". This was very unclear to me; and an analytical example or empirical evaluation of this is necessary; otherwise it should be cut.
- label the axes in the figures
- align the algorithm boxes
- The authors state "the name is misleading" about variational thompson sampling. Please use a different word than "misleading" or rename the paper/framework.

Significance:
- as above, I like the theoretical perspective this paper presents, and think if it is made more accessible through the use of visual aids, diagrams, and more experiments, it will result in practical impact as well.

Edit:
- have updated the score based on the strength of the author's response. Thank you for taking the time!

**Time Spent Reviewing:**

2

---

> ### Author Response · Authors · 2021-08-09
> **Reponse to reviewer AmXN**
>
> Thank you for your comments and feedback on the manuscript!
>
> * “significant revision in terms of clarity."
>
>  We have made some significant changes to the text based on your suggestions. In particular, many sentences have been rewritten to make the noun more central.
>
> * “other 'walls of text”
>
> The walls of text have been expanded and clarified, this can be done now because the camera ready version allows an extra page, which yields more space for discussion and spacing out the equations.
>
> * “a visual aid of Gμ would go a long way”
>
> Is Figure 2 sufficient? If not, is there a particular figure in Wainwright and Jordan that would make it clearer?
>
> * "it permits the use of TS in a differential computation framework, such as a layer in a neural network".
>
> We have expanded on and clarified what we meant by that. In short, the VTS objective is differentiable and concave and therefore can be optimized via gradient descent to find the VTS policy, which may be done as part of a neural network layer using backpropagation. No such variational form of (pure) TS exists to our knowledge.
>
> * “label the axes in the figures...align the algorithm boxes”
>
> We have added labels to the figures and tried to align the algorithm boxes to be more visually pleasing and consistent.
>
> * “rename the paper/framework.”
>
> Do you think VOBS (Variational Optimistic Bayesian Sampling) is a better name?

---

> ### Comment · Reviewer_AmXN · 2021-08-11
> **Updated score**
>
> > Has the author response adequately addressed any of your concerns?
>
> Yes.
>
> > Do the other positive reviews change your view of the paper?
>
> Yes.
>
> > Your critique centered mainly on the clarity of the draft. Is the poor clarity the sole reason for your rating of 4, or do you have concerns with the substance of the paper?
>
> Yes, it was mainly the clarity and I am glad the author's are taking steps to remedy this (such as by considering alternative names that are more precise!).
>
> > Do you think VOBS (Variational Optimistic Bayesian Sampling) is a better name?
>
> Yes.
>
> > Is Figure 2 sufficient? If not, is there a particular figure in Wainwright and Jordan that would make it clearer?
>
> Figure 2 is helpful. Not sure, but it might be possible to make a cartoon illustration of the equation below L129 in the submission.
> Figure 5.3 (the cartoon), 3.5, 4.2 in https://people.eecs.berkeley.edu/~wainwrig/Papers/WaiJor08_FTML.pdf are examples of what I was thinking of.

---

> > ### Author Response · Authors · 2021-08-13
> > **Reponse to update from reviewer AmXN**
> >
> > Thank you for your comments and for updating the review!
> >
> > On the figure: Fig 2 is essentially trying to demonstrate that equation (below L129) by plotting the set of all policies that satisfy the inequality (in blue) and adding the TS policy and the VOBS policy as specific points. It's also trying to show how the set contracts with more data. We are also adding another figure similar to Fig 2 to the manuscript to deal with the bilinear case showing that TS can leave the shaded set in some cases. Also we have improved the readability (larger labels etc.) and the caption. Hopefully this is enough to make it clear to the reader what that figure is trying to convey.

---

### Official Review · Reviewer_2rdU · 2021-07-19

**Rating:** 7
**Confidence:** 3

**Summary:**

This paper introduces Variational Thompson Sampling, where rather than sampling from the posterior, which can be infeasible, a computationally tractable upper bound is optimized.  In the stochastic multi-armed bandit setting with subgaussian noise, they provide a general analysis for any algorithm that plays a "stochastically optimistic" policy, which includes vanilla TS and the new method.  The VTS algorithm is shown to have sublinear regret for the more challenging bilinear saddle point problems where it can be shown that TS fails.

**Limitations And Societal Impact:**



**Main Review:**

The paper seems technically sound.  The introduction of stochastically optimistic policy set is a useful abstraction that provides a nice joint analysis of TS and VTS and is useful for understanding.

The algorithm and analysis seem novel, but from a theoretical standpoint, it does not seem to improve over existing algorithms.  Experimentally, it seem marginally better than the similar K-learning algorithm.  The fact that the VTS objective is differentiable and could be incorporated as a NN layer is an interesting idea.

The paper is well written.  In particular, the paper does a good job of relating to other algorithms and interpretation in Section 3.

As noted in the paper, the name variational TS is a bit misleading.  While a better name may be possible, the clarification is appreciated.

In Section 4 it is noted that TS can suffer linear regret in the zero-sum game setting because it is open to exploittation without revealing information about the reward matrix. Some high level intuition on how VTS avoids this would be useful here.



Minor comments:
Line 90.5: $\mu$ should be $X$.
Line 235: Should $\Delta_A$ be replaced with $\Delta_n$?

**Time Spent Reviewing:**

4

---

> ### Author Response · Authors · 2021-08-09
> **Response to 2rdU**
>
> Thank you for your feedback and suggestions!
>
> “Experimentally, it seem marginally better than the similar K-learning algorithm”.
>
> K-learning and VTS are similar, but differ in that VTS provides the ability for each arm to have its own ‘temperature’ parameter, whereas for K-learning they all must share the same temperature. For problems with more arms this difference can be significant. So, with this in mind, we have re-run all the experiments with more arms to demonstrate a more significant performance improvement over K-learning (initial experiments were with small numbers of arms).
>
> “the name variational TS is a bit misleading”
>
> We agree, do you think VOBS (Variational Optimistic Bayesian Sampling) is a better name?
>
> “In Section 4 it is noted that TS can suffer linear regret in the zero-sum game setting because it is open to exploittation without revealing information about the reward matrix. Some high level intuition on how VTS avoids this would be useful here.”
>
> We have expanded on this in the text. In short the difference is that VTS is guaranteed to be _optimistic_ about every strategy (since it is constructing upper bounds on each entry). Since it is optimistic, exploiting it would reveal information to it. On the other hand Thompson sampling is sometimes pessimistic, and in those cases it can be exploited without revealing information.
>
> “Minor comments: Line 90.5: μ should be X. Line 235: Should ΔA be replaced with Δn?”
>
> You are right, thank you. These have been fixed.

---

> > ### Comment · Reviewer_2rdU · 2021-08-25
> > **Quick response**
> >
> > Thanks for your helpful explanations regarding K-learning and the intuition on optimism in VTS vs TS.  I think VOBS would be a better name.

---

### Author Response · Authors · 2021-08-09
**Thank you to the reviewers**

We thank the reviewers for their careful reading of the paper and their insightful feedback. We respond to each reviewer below. We would be grateful if the reviewers could take a look at the responses and adjust the scores if we have addressed their concerns appropriately. If we have missed anything please let us know.

One point brought up by several reviewers was the name ‘Variational Thompson sampling’. We agree that this name is not perfect (naming is one of the hardest problems in computer science!). We propose to change the name (and title) to ‘Variational Optimistic Bayesian Sampling (VOBS)’. Does this sound reasonable to the reviewers?

As well as the improvements to the paper listed below, we also re-ran all the numerical experiments with more ‘arms’ - more arms makes the problem harder and it better shows how VTS outperforms K-learning. We also derived the ‘stochastically optimistic set of policies’ for the bilinear case (which was missing before) and added a figure similar to Fig 2 in the paper but for the bilinear case showing that TS can actually  _leave_ the stochastically optimistic set for bilinear saddle-point problems.

---

### Decision · Program_Chairs · 2021-09-27

**Decision:**

Accept (Spotlight)

**Comment:**

This paper introduces a family of stochastically optimistic policies that achieve favourable regret. All of the reviewers were enthusiastic about this paper, noting that the contribution was "technically sound" and "elegant".  I agree it is a valuable contribution, and will therefore recommend acceptance. I want to note two concerns that the authors should make sure to address in the camera ready:
* Please de-emphasize the argument that sampling from the posterior is harder than computing integrals w.r.t. the posterior. Unless I've misunderstood, I'm not sure I agree with this argument.
* Multiple reviewers recommended that the name of the method be changed. While I cannot enforce this, I also believe it is in the best interest of the authors to pick a name that the community finds appropriate.